# Assessing Health-Related Quality of Life in Children with Coeliac Disease: The Italian Version of CDDUX

**DOI:** 10.3390/nu13020485

**Published:** 2021-02-02

**Authors:** Marco Crocco, Angela Calvi, Paolo Gandullia, Federica Malerba, Anthea Mariani, Sonia Di Profio, Barbara Tappino, Stefano Bonassi

**Affiliations:** 1Department of Pediatrics, IRCCS Giannina Gaslini Institute, 16147 Genova, Italy; federica-malerba@libero.it; 2Department of Neuroscience, Rehabilitation, Ophthalmology, Genetics, Child and Maternal Health, University of Genova, 16132 Genova, Italy; 3Gastroenterology and Digestive Endoscopy Unit, IRCCS Giannina Gaslini Institute, 16147 Genova, Italy; angelacalvi@gaslini.org (A.C.); paologandullia@gaslini.org (P.G.); 4Department of Pediatrics, Ospedale Santo Spirito, 65124 Pescara, Italy; antheamariani@gmail.com; 5Psychology Unit, IRCCS Giannina Gaslini Institute, 16147 Genova, Italy; soniadiprofio@gaslini.org; 6LABSIEM (Laboratory for the Study of Inborn Errors of Metabolism), IRCCS Giannina Gaslini Institute, 16147 Genova, Italy; barbaratappino@gaslini.org; 7Department of Human Sciences and Quality of Life Promotion, San Raffaele University, 00166 Rome, Italy; stefano.bonassi@sanraffaele.it; 8Unit of Clinical and Molecular Epidemiology, IRCCS San Raffaele Pisana, 00166 Rome, Italy

**Keywords:** coeliac disease, Health-Related Quality of Life, disease-specific questionnaire, gluten free diet, children, CDDUX

## Abstract

We aimed to assess Health-Related Quality of Life (HRQoL) of Italian children and their parents with coeliac disease (CD) using the Coeliac Disease Dutch Questionnaire (CDDUX). The CDDUX underwent a cross-cultural adaptation in a multi-step process, according to international guidelines. A total of 224 children aged between 8–18 years and their parents were prospectively recruited. Cronbach α coefficient was determined as a measure of internal consistency of the questionnaire and inter-children/parent reliability by intraclass correlation coefficient. Univariate and bivariate regression models were used to evaluate correlations between clinical variables and children and parents subclasses of CDDUX and overall mean Paediatric Quality of Life Inventory (PedsQL). The Italian CDDUX proved to be valid and reliable, mean CDDUX total score revealing a neutral evaluation of the quality of life in children 52.6 ± 17.2 and parents 49.5 ± 17.9 (*p* = 0.07) with strong correlation with PedsQL. The only clinical variable which appeared to affect significantly quality of life both in children and parents was the lower age. A comparison with our results showed remarkable differences in the HRQoL of populations of various nationalities. The Italian version of the CDDUX questionnaire is a simple and reliable tool for assessing the HRQoL in children and adolescents with CD.

## 1. Introduction

Coeliac disease (CD) is a life-long multiorgan immune-mediated disease triggered by ingestion of gluten in genetically susceptible individuals. Symptoms are heterogeneous and may interest a various range of gastrointestinal and non-gastrointestinal districts at different ages [1,2].

To date, the only treatment available is a gluten-free diet (GFD). This approach requires significant compliance from young patients and may be difficult to follow given the major changes to eating habit, lifestyle, and the life-long duration [3]. The need to follow a GFD, associated to the chronic trait of illness, could cause stigma in society and a consequent lower quality of life [4]. Assessing the impact of GFD on the quality of life (QoL) of paediatric patients and their parents is, therefore, a priority, to be addressed with validated tools [5].

Health-Related Quality of Life (HRQoL) is a multidimensional concept, which aims to measure well-being, taking into account subjective attitudes, and experiences from physical, emotional, social and cognitive domains [6]. The evaluation of HRQoL in coeliac patients is often conducted using non-specific questionnaires [7]. Recently, disease-specific questionnaires have been developed for CD, but only few of them have been specifically targeted for children. Kolestern and colleagues [8] were the first to propose disease-specific questions; however, the Technisch Natuurkundig Onderzoek Academisch Ziekenhuis Leiden Children’s quality of life—Coeliac disease–questionnaire (TAQOL-COE), is a tool based on symptomatic questions and does not provide information about the children’s view [9]. The Coeliac Disease Dutch questionnaire (CDDUX), developed by van Doorn in 2009, was the first validated disease-specific HRQoL questionnaire for children (age 8 to 18) with CD and their parents [9,10]. Although Jordan and colleagues in 2013 validated another questionnaire for U.S. children and adolescents [11], the CDDUX had better performances, and in these years, it has been translated and adapted for several non-English speaking countries [12,13,14,15]. Currently, no CD-specific questionnaire is available in Italian to assess the HRQoL in children and adolescents. The lack of a HRQoL assessment specific tool validated for use in Italy did not allow to compare psychometric characteristics of Italian families in an international context. The aim of the present study was to assess the HRQoL in children and adolescents with CD and their parents, using a version of the CDDUX questionnaire translated and adapted to Italian coeliac families. These results will be compared to a generic tool such as the Paediatric Quality of Life Inventory (PedsQL) Mapi Research Trust, (Lyon, France) generic core v4.0 [16].

## 2. Materials and Methods

### 2.1. Assessment of HRQoL and a Cross-Cultural Adaptation of the CDDUX Questionnaire

The CDDUX questionnaire is composed by 12 items on three subscales, focused on three different domains of possible daily experiences, i.e., “Diet” (six items), where the child is invited to express their feelings about compliance, restrictions of the GFD, and the lifelong aspects; “Communication” (three items), where the child is invited to express their feelings when talking about CD to others and when explaining the disease to others; “Having CD” (three items), where the child is invited to express their feelings when offered food containing gluten or what they think about food containing gluten. The CDDUX, in addition to the version for children (8–18 years), has a version for parents which assesses the influence of the disease on children’s daily activities, in the three domains described above. The answers in both questionnaires are marked on a five-point Likert face scale. Each face corresponds to a score, increasing up to a value of 100. The HRQoL is considered as very bad for scores up to 20, while 21–40 is bad, 41–60 is neutral, 61–80 is good, and 81–100 is very good.

The PedsQL is a generic tool specifically developed to evaluate the quality of life in chronically ill children. Questionnaires for children 8–12 and adolescents 13–18 years, and proxy versions of the questionnaire, are already available in Italian. PedsQL is composed of 23 items with four subscales, focused on different functioning domains: Physical (eight items), Emotional (five items), Social (five items), and School (five items).

Both questionnaires (Italian version CDDUX and PedsQL) were self-completed independently by the children and their parents, at the end of the clinical visit, after a brief explanation.

To use the CDDUX with Italian children, the questionnaire was cross-culturally adapted in a multi-step process, according to international consensus guidelines [17], and with the following steps: 1—forward translation, from the original English versions into the Italian version (translation agreed upon by a medical doctor, a paediatric gastroenterologist, and a coeliac patient, all of them fluent in English); 2—backward translation, from the first Italian version into English (by an independent paediatric gastroenterologist, a medical doctor expert in the coeliac field, and a different coeliac patient); 3—test and retest the reliability, where all those involved in the translations met together to compare the versions and reconcile the differences; after the consensus, the final Italian version was tested with a group of five patients and their parents to check the appropriateness and clarity of the questionnaire. Lastly, a random sample of five patients and their parents was re-contacted by email and phone after 7 days for re-testing the reliability. The final Italian version was validated by analysing its psychometric properties, including reliability and validity.

### 2.2. Patients

The Italian version of CDDUX and PedsQL generic core 4.0 were administered consecutively, to all children 8–18 years examined for 12 months at our centre, and their parents. A total of 224 families out of 234 (95.7%) were willing to participate to the study. One family did not fully complete the questionnaire and was, therefore, excluded from the analysis. All patients were followed up by the coeliac day-hospital of the regional centre for coeliac disease at the children hospital Giannina Gaslini, Genoa, Italy. All diagnoses were confirmed according to the clinical criteria of European Society for Paediatric Gastroenterology Hepatology and Nutrition (ESPGHAN) [18] or by histological damage in duodenal biopsy samples. Demographic and clinical parameters (sex, age, time since diagnosis, family history of CD, associated diseases, result of the last anti-transglutaminase test), possibly associated with HRQoL, were included in the questionnaires. When variables were not available for some patients, these were excluded for percentage calculation (Table 1). Subjects affected by psychiatric disorders, cancer, or with poor knowledge of Italian language were excluded from the study. All parents signed an informed consent for inclusion in the study. The study was conducted in accordance with the Declaration of Helsinki, and the protocol was approved by the Regional Ethics Committee (protocol code 132/19).

### 2.3. Statistical Analysis

Demographic and clinical data were expressed as mean and standard deviation (SD) for continuous variables and as proportions for categorical variables. Distribution frequencies were estimated for qualitative variables, and scores for the different scales are reported as mean with their SD. Patients were stratified by age as children (8–11 years), adolescents (12–15 years), and young adult (16–18 years), or by time since diagnosis (1–4, 5–8, and 9–16 years).

The psychometric properties of the tool were analysed by use of classic validation methods, i.e., intra-questionnaire reliability by Cronbach’s α coefficient and inter-children/parent reliability by intraclass correlation coefficient and 95% confidence intervals (CI) within child/parent pairs of CD patients by total and by classification indexes of the Italian version of CDDUX. By international agreement, internal consistency is indicated as: 1 < α ≥ 0.9 excellent, 0.7 ≤ α < 0.9 good, 0.6 ≤ α < 0.7 acceptable, 0.5 ≤ α < 0.6 poor, and α < 0.5 unacceptable [19]; reliability value is indicated as: 1–0.8 indicates excellent, >0.6 good, >0.4 moderate, and <0.4 poor [20]. Student’s *t*-test was used to compare the mean score of the questionnaires administered to the children and their parents; the same test was also used to compare the differences in the means obtained by the various authors in the different countries where the CDDUX was successfully validated. Eventually, the influence of demographic and clinical features on the quality of life, measured with the CDDUX and the PedsQL, was evaluated with a univariate analysis based on the student’s t-test for independent samples. All data were analysed with the software SPSS for Windows (IBM SPSS Statistics for Windows, v26. IBM Corp., Armonk, NY, USA).

## 3. Results

### 3.1. Study Population

The CDDUX questionnaire was proposed to 234 families, and 224 of them (95.7%) accepted to participate to the study. The questionnaire was fully completed by 223 children (99.6%) and 219 parents (97.8%), while the PedsQL were completed by 216 children and parents (96.4%). Clinical data of 23 children were not available. The proportion of females was higher (64.1% vs. 35.9%), with an age equally distributed between 8 and 18 years. The duration of the gluten-free diet (GFD) was re-classified in three subgroups: <4; 5–8; >9 years. In CD patients, mean age at diagnosis was 12.6 years (SD ± 3.0, range 8–18), with a total of 78 children (39.6%) who reported a family history for CD. Demographic and clinical features of the 223 participating pairs are reported in Table 1. At the time of enrolment, 23 patients (12%) showed a recent high value of anti-tissue transglutaminase antibodies (TTG), a possible index of a not well-controlled GFD. Coeliac disease was associated in 44 children (21.9%) to other chronic diseases; specifically, they were affected by autoimmune thyroiditis (10), allergies (8), asthma (5), atopic dermatitis (4), type 1 diabetes mellitus (2), short stature (2), other diseases (13) (four cases of isolated hyperthyrotropinemia, and one case each of impaired glucose tolerance, alopecia, common variable immunodeficiency, eosinophilic gastroenteritis, psoriasis, congenital heart disease, TIC disorder, gallstones, hypertriglyceridemia).

### 3.2. HRQoL and Its Related Clinical Variables

To evaluate the agreement between children and parents concerning HRQoL, mean scores of the CDDUX and PedsQL were compared between the two of them. Mean CDDUX total score was 52.6 ± 17.2 in the group of children and 49.5 ± 17.9 in the group of parents (*p* = 0.07). Most results were included between 40 and 60, revealing a neutral evaluation of the quality of life, and in most cases these scores did not significantly differ between children and their parents (Table 2).

The only answers which showed significantly different values is the way of communicating to other persons the presence of CD, which is apparently much easier for children (62.4 vs. 57.9; *p* < 0.05). Mean total PedsQL score was 81.2 ± 11.1 and 80.3 ± 12.9 in children and parents, respectively, with no differences detected in paired comparisons between the two groups (Table 3), except in the dimension social functioning, where children were shown to find it easier to approach their problem (89.2 vs. 84.3; *p* < 0.01). The overall mean score reported for the CDDUX was eventually compared with the already-validated PedsQL questionnaire (Figure 1a,b), showing significant correlations both in children (*r* = 0.302, *p* < 0.001) and to a lower extent in parents (*r* = 0.230, *p* < 0.01).

### 3.3. Psychometric Properties and Validation of the Italian CDDUX Questionnaire.

The internal consistency of the CDDUX questionnaire and the degree of correlation between the tools for children and parents were evaluated. The bivariate correlation analysis reported in Table 4 showed correlation coefficients ranging between +0.196 and +0.494. All coefficients were statistically significant (*p* < 0.01).

The Cronbach’s α coefficient showed a high or excellent internal consistency for the overall CDDUX tool, while a lower consistency was observed—both for children and parents—in the subclass of answers describing children’s feelings about living with the disease in society, i.e., 0.68 in both groups. Similarly, the intraclass correlation coefficient revealed a significant degree of correlation (*p* < 0.001) for all subclasses (Table 5).

Table 6 and Table 7 report the association of demographic and clinical variables with the CDDUX (overall and subclasses) and the PedsQL (overall), with the aim of ranking those conditions which affect the quality of life according to children and parents. The only parameter which appeared to significantly affect quality of life according to children was age, with coeliac children in the age 8–11 years reporting a lower mean in the CDDUX ‘Having coeliac disease’ score (36.5 ± 19.5; *p* < 0.05) and in the overall PedsQL score (77.1 ± 12.3; *p* < 0.05) (Table 6). On the other hand, parents reported more conditions affecting the quality of life according to the CDDUX questionnaire. Lower scores were reported for children with younger age, i.e., 8–11 years (34.6 ± 19.3, *p* <0.01), positive to TTG (32.1 ± 17.6; *p* < 0.05), and with a recent diagnosis of CD, i.e., 1–4 years (33.1 ± 19.1; *p* < 0.01). No association was found in the parents score between the conditions considered and the PedsQL scores. 

Finally, a comparison was made between the CDDUX results of the various authors, obtained in children with CD and their parents, all of which described populations with different nationalities (Figure 2).

## 4. Discussion

GFD is the only available treatment for children with CD, and the compliance to this diet could be severely influenced by the quality of life. Only a few studies investigated HRQoL in coeliac children, by using different tools and reporting conflicting results [21]. HRQoL was assessed using the qualitative tool (open-ended questions) [21] of generic questionnaires [16,22]; in some studies, CD-specific questions exploring wellbeing and lifestyle were included [23,24]. To date, only two paediatric disease-specific HRQoL tools have been validated to assess the QoL in children with CD: the CDDUX available in Dutch and English, later adapted into Spanish, Argentine, Portuguese, and Iranian [9,12,13,14,15], and the CDPQoL, validated in English for U.S. patients. In this study, we adapted the English version of CDDUX for use in Italian children, preserving its concepts and the equivalence of contents. Cross-cultural adaptation, following a multi-step process, was performed satisfactorily. There was a consensus to replace the term “find/trovare” with “feel/sentire” in the adapted final version in order to present questions from an emotional prospective. As expected, there is a significant difference between the two sexes, with females accounting for 64.1% of the population.

The results of our study confirm that CDDUX has excellent psychometric properties, including a good internal consistency and strong correlation with generic QoL instrument (PedsQL). However, PedsQL may fail to detect the discomfort of children and parents regarding communication and nutrition in their everyday life due to their condition. The CDDUX, by providing information on three subscales focused on the specific aspects of daily life, allows to appreciate with greater accuracy what children and parents think and feel about CD, and the fears and limitations related to a life-long gluten-free diet. Statistical analysis showed that the main factors related to having a worse HRQoL with CDDUX were lower age at interview, scarce adherence to following GFD, and a recent diagnosis. 

We stress that studies performed in the various national states are representative of specific recruiting regions, and therefore, automatic generalisation to the national population of other regions may not be reliable. Although considering this limit, in this case, children with CD and their parents showed similar HRQoL to Portuguese families [13], but apparently better than Dutch and Iranian [9,15] and worse than Spanish, Argentines, and Chileans [12,14,25] (Figure 2).

A critical strength of our study is the high enrolment rate. Most families were very collaborative, and the number of families who did not accept the invitation to participate to the study was minimal. As regards sample selection, there was the advantage of using outpatients from a large clinic in the only children hospital of the region; therefore, no preconditions were selected that may have affected the answers, such as when using patient associations. On the other hand, our cases came from the Gaslini Hospital, which is a tertiary referral centre, and all the families came from a single Italian region (Liguria). Although our findings cannot be generalised to the whole population of Italian children and adolescents with CD, no evident selection biases affected our results. Another possible limitation is the absence of a healthy control group, and the lack of data on the socio economics status of families. The use of a disease-specific HRQoL questionnaire could bring out problems that a patient may keep to themself in the anamnestic interview. The questionnaire allows patients to express problems directly and without fear of judgment by parent or doctor, revealing difficulties with the GFD, thus allowing to improve the compliance. Further multicentre studies with larger populations, representative of the regional multiculturalism, are necessary for a better understanding of the impact of coeliac disease on the quality of life in Italian children and adolescents.

## 5. Conclusions

The findings of our study suggest that HRQoL should be investigated routinely in the follow-up of the paediatric patient with a questionnaire tailored to assess the peculiarities of coeliac disease. The Italian version of the CDDUX questionnaire seems to be a simple, quick, complete, and promising tool for assessing HRQoL in children and adolescents with CD.

## Figures and Tables

**Figure 1 nutrients-13-00485-f001:**
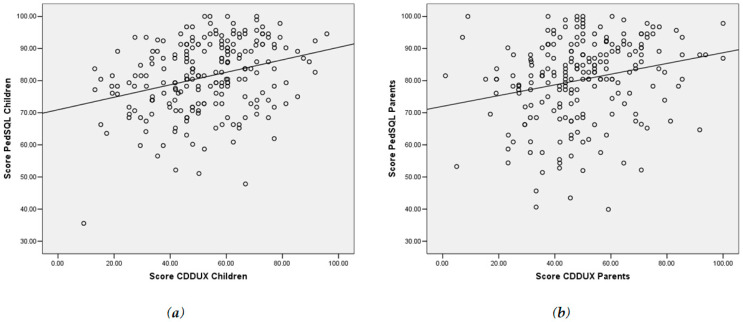
Scatter plot of the PedsQL overall mean score and the CDDUX overall mean score. (**a**) Children *r* = 0.302, *p* < 0.001; (**b**) Parents *r* = 0.230, *p* < 0.01.

**Figure 2 nutrients-13-00485-f002:**
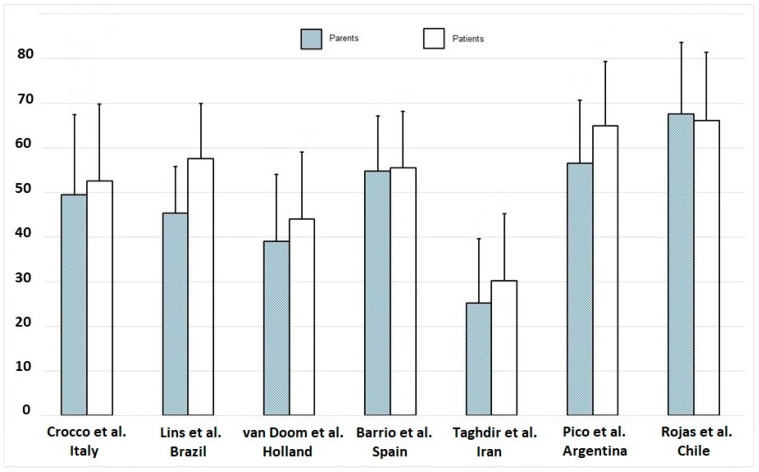
CDDUX mean and SD obtained by coeliac patients and parents in different countries.

**Table 1 nutrients-13-00485-t001:** Demographic and clinical characteristics of coeliac disease (CD) patients.

Demographic and Clinical Characteristics	*n*	%
	223	(100%)
Sex		
Male	80	(35.9%)
Female	143	(64.1%)
Age at interview		
8–11 y	47	(32.2%)
12–15 y	58	(39.7%)
16–18 y	41	(28.1%)
TTG		
Positive	23	(12.0%)
Negative	169	(88.0%)
Time since diagnosis (years)		
1–4 y	41	(24.6%)
5–8 y	78	(46.7%)
9–16 y	48	(28.7%)
Family history of CD		
Positive	78	(39.6%)
Negative	119	(60.4%)
Relative affected by CD ^1^		
Grandfather/Mother	5	(6.0%)
Father	12	(14.2%)
Mother	31	(36.9%)
Brother	11	(13.1%)
Sister	14	(16.7%)
Others	11	(13.1%)
Associated diseases		
Yes	44	(21.9%)
No	157	(78.1%)
Most common associated diseases		
Thyroiditis	10	
Allergies	8	
Asthma	5	
Dermatitis	4	
DM I	2	

^1^ Nine patients reported two CD cases within first-degree relatives; three patients reported three CD cases among first-degree relatives. Coeliac disease (CD); years (y); diabetes mellitus (DM I).

**Table 2 nutrients-13-00485-t002:** Assessment of Health-Related Quality of Life (HRQoL) in the 223 child/parent pairs of CD patients according to the Italian version of Coeliac Disease Dutch Questionnaire (CDDUX).

CDDUX	*n*	Mean	SD	Mean Difference	*p* Value
Total					
Children	223	52.6	17.2	3.04	
Parents	219	49.5	17.9		0.07
Having coeliac disease					
Children	223	42.8	21.1	1.51	
Parents	219	41.3	19.8		n.s.
Communication					
Children	223	62.4	22.0	5.12	
Parents	219	57.3	22.2		<0.05
Diet					
Children	223	48.4	21.7	1.53	
Parents	219	46.9	21.5		n.s.

Not significant (n.s.); Standard deviation (SD).

**Table 3 nutrients-13-00485-t003:** Assessment of HRQoL in the 216 child/parent pairs of CD patients according to the Italian version of the Paediatric Quality of Life Inventory (PedsQL).

PedsQL	Mean	SD	Mean Difference	*p* Value
Children	81.2	11.1	0.91	
Parents	80.3	12.9		n.s.
Physical functioning				
Children	87.7	12.5	3.19	
Parents	84.5	16.3		n.s.
Emotional functioning				
Children	73.7	16.5	2.18	
Parents	71.6	17.0		n.s.
Social functioning				
Children	89.2	14.4	4.88	
Parents	84.3	18.3		<0.01
School functioning				
Children Parents	76.7 77.3	16.0 19.0	−0.59	n.s.

Not significant (n.s.).

**Table 4 nutrients-13-00485-t004:** Bivariate correlation analysis between child/parent pairs of CD patients by subclasses indexes of the Italian version of CDDUX.

Subclasses Indexes of CDDUX		Having Coeliac Disease	Communication	Diet	Total
		Parents	Parents	Parents	Parents
**Having coeliac disease**	Children	0.359 **	0.225 **	0.287 **	0.342 **
**Communication**	Children	0.196 *	0.379 **	0.295 **	0.354 **
**Diet**	Children	0.253 **	0.322 **	0.392 **	0.413 **
**Total**	Children	0.342 **	0.359 **	0.416 **	0.494 **

* *p* < 0.01; ** *p* < 0.001.

**Table 5 nutrients-13-00485-t005:** Intra-questionnaire reliability (Cronbach’s α coefficient) and inter-children/parent reliability (intraclass correlation coefficient) between child/parent pairs of CD patients by total and subclasses indexes of the Italian version of CDDUX.

Subclasses Indexes	Parents	Children		
CDDUX Score	Cronbach’s α	Cronbach’s α	Intraclass Correlation Coefficient (95% CI)	*p* Values
**Having coeliac disease**	0.682	0.681	0.298 (0.241–0.360)	*p* < 0.001
**Communication**	0.869	0.788	0.416 (0.357–0.846)	*p* < 0.001
**Diet**	0.936	0.873	0.421 (0.371–0.476)	*p* < 0.001
**Total**	0.938	0.881	0.332 (0.289–0.381)	*p* < 0.001

**Table 6 nutrients-13-00485-t006:** Association of demographic and clinical selected variable with overall mean PedsQL and overall and subclasses of CDDUX (Children).

Children		CDDUX				PedsQL
Parameter	*n*	Having Coeliac Disease	Communication	Diet	Total	Total
Sex						
Male	80	43.7 (21.2)	62.2 (20.7)	47.2 (21.1)	52.2 (15.8)	80.7 (11.6)
Female	143	42.4 (21.1)	62.3 (22.6)	48.9 (22.2)	52.7 (18.0)	81.4 (10.9)
Age at interview						
8–11 y	47	36.5 (19.5) *	62.3 (24.2)	50.5 (20.3)	51.3 (17.0)	77.1 (12.3) *
12–15 y	58	44.4 (22.4)	64.2 (21.0)	48.3 (20.7)	54.2 (16.7)	83.1 (9.6)
16–18 y	41	48.3 (18.2)	62.2 (21.1)	47.4 (19.3)	51.3 (16.4)	82.4 (9.6)
TTG						
Positive	23	38.6 (20.0)	59.9 (23.8)	49.2 (21.1)	52.0 (16.4)	80.6 (10.9)
Negative	169	42.6 (22.0)	62.9 (21.7)	49.5 (21.1)	52.9 (17.1)	80.5 (11.4)
Time since diagnosis (years)						
1–4 y	41	42.6 (20.4)	65.5 (20.2)	53.1 (16.9)	55.1 (12.6)	82.6 (11.3)
5–8 y	78	42.3 (21.3)	61.3 (22.3)	49.4 (20.5)	52.5 (17.2)	79.5 (11.9)
9–16 y	48	48.4 (21.9)	65.3 (19.5)	51.7 21.6)	54.9 (18.3)	82.9 (10.4)
Family history of CD						
Positive	78	42.2 (22.1)	62.1 (21.7)	48.5 (20.6)	51.5 (18.3)	79.3 (11.5)
Negative	119	42.4 (21.5)	62.3 (22.3)	49.6 (21.4)	53.2 (16.3)	81.7 (11.0)
Associated diseases						
Yes	44	41.6 (23.3)	63.9 (25.5)	49.2 (22.2)	52.6 (17.9)	78.2 (14.4)
No	157	42.7 (21.1)	62.0 (20.9)	49.2 (20.8)	52.6 (16.8)	81.3 (10.1)

* *p* < 0.05.

**Table 7 nutrients-13-00485-t007:** Association of demographic and clinical selected variable with overall mean PedsQL and overall and subclasses of CDDUX (Parents).

Parents		CDDUX				PedSQL
Parameter	*n*	Having Coeliac Disease	Communication	Diet	Total	Total
Sex						
Male	80	40.5 (18.3)	56.2 (23.1)	46.6 (22.1)	49.2 (17.7)	79.8 (14.8)
Female	143	41.7 (20.7)	57.8 (21.8)	47.0 (21.4)	49.7 (18.2)	80.5 (11.9)
Age at interview						
8–11 y	47	34.6 (19.3) **	56.4 (21.0)	44.1 (21.4)	45.9 (17.9)	78.4 (13.6)
12–15 y	58	44.2 (18.9)	55.8 (22.1)	46.6 (22.8)	50.0 (18.2)	82.7 (11.0)
16–18 y	41	46.9 (20.4)	60.4 (22.7)	50.5 (20.9)	52.2 (19.1)	82.0 (11.9)
TTG						
Positive	23	32.1 (17.6) *	50.1 (26.3)	40.7 (22.0)	45.3 (17.8)	77.6 (15.3)
Negative	169	42.2 (19.7)	58.2 (21.6)	47.8 (21.0)	50.0 (17.8)	80.5 (12.8)
Time since diagnosis (years)						
1–4 y	41	33.1 (19.1) **	52.3 (22.7)	42.3 (22.1)	46.8 (16.8)	76.9 (14.9)
5–8 y	78	42.3 (19.2)	57.8 (19.9)	47.1 (18.4)	49.0 (16.6)	81.1 (12.2)
9–16 y	48	46.3 (19.6)	58.0 (25.9)	50.1 (23.8)	52.2 (19.9)	83.4 (13.2)
Family history of CD						
Positive	78	40.3 (21.0)	56.6 (24.1)	46.7 (23.6)	49.0 (20.0)	79.1 (14.4)
Negative	119	41.9 (19.2)	57.7 (21.7)	47.6 (19.8)	50.1 (16.8)	80.5 (12.2)
Associated diseases						
Yes	44	40.5 (15.6)	55.7 (22.4)	45.4 (19.2)	47.4 (16.1)	77.7 (15.1)
No	157	41.2 (21.0)	57.7 (22.5)	47.4 (21.8)	50.0 (18.5)	80.6 (12.3)

* *p* < 0.05; ** *p* < 0.01.

## Data Availability

The data presented in this study are available on request from the corresponding author.

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
