# Peer review of "Assessing Health-Related Quality of Life in Children with Coeliac Disease: The Italian Version of CDDUX"

_nutrients, 2021, doi:10.3390/nu13020485_

Round 1
Reviewer 1 Report
Crocco et al. reported an evaluation study in which they modified the CDDUX for Italian patients and examined its reliability and validity among 224 child-parent dyads. Their study process was clearly described in the manuscript.
There are two points that need to be further explained and clarified in order to better inform practitioners of the clinical relevance of this questionnaire.
- How were the forms (Italian version CDDUX and PedsQL) administered? Were they administered by an interviewer or self-completed by the children and their parents? The authors pointed out in the discussion the possibility of respondents holding thoughts to themselves, therefore it is critical to know whether interviewers were involved in data collection.
- The correlation coefficients presented in Table 4 were statistically significant, however the values did not suggest a strong correlation according to the criteria cited by the authors in the Statistical Analysis section. Have the authors compared the correlation coefficients by “time since diagnosis” and other variables presented in Table 1?
There are a few occasional typos throughout the article that warrants proofreading.
Reviewer 2 Report
Assessing Health-Related Quality of Life in children with coeliac disease: the Italian version of CDDUX
This is a well-conducted case series study of celiac patients and parents to construct and validate and Italian version of the CDDUX. Some interesting preliminary findings are discussed and compared to other CDDUX findings from other countries. There are no major issues with the design or presentation of the study. Minor issues include a careful look at abbreviations and at English usage.
Abstract
There is a certain amount of carelessness in the Abstract concerning consistent spelling and abbreviations. Celiac is also spelled Coeliac. HRQoL is also spelled HRQOL. PedSQL is not defined. PedSQL is also spelled PedsQL. With such a large standard deviation “52.57 ± 17.17” could be “52.6 ± 17.2” or even “53 ± 17” without losing any meaning. “49.53 ± 17.9” should have the same number of significant figures and suffers from the same large standard deviation. Is age a clinical parameter?
Introduction
Is it coeliac or celiac? Please be consistent.
The use of English in the Introduction is noticeably awkward, especially compared to other sections of the paper. I suggest a very careful re-reading of this section. The use of shorter sentences would help in these awkward situations.
Results
The number of significant figures in Tables 2 and 3 are more than in Tables 6 and 7. Tables 2 and 3 have more than is needed based on the large standard deviations.
Discussion
The significance indicators in Figure 2 are confusing. The caption does not describe the blue and white bars. I am assuming that one are parents and one are children. If the marks are meant to indicate the difference between bars, how can the significance be different? If parents are different from children at p<0.001 then how can children be different from parents at p<0.05? It appears that the first and third mark both mean p<0.001. Why does the first blue bar have 2 symbols? A better caption might help.
Throughout the paper better attention is needed to abbreviations such as HRQOL and HRQoL.
Among the limitations is this statement: “Another possible limitation is the absence of a healthy control group…” However, in the Methods we find this statement, “To compare the value of HRQoL in celiac children with a healthy reference groups, we used the PedsQL.” I could not find any Results that reflected a healthy reference group. Should this section of the Methods be removed? This is confusing.
In the paragraph starting on line 255, that compares the present study and CDDUX results with those from other studies, it may be useful to remind readers that those studies probably also suffer from the same lack of national representativeness that the current study does. How can a local study of Iranians compare to a local study of Italians? Answer-Very cautiously.
